# Unplanned Resections of Soft Tissue Sarcomas—Necessity of Re-Resection?

**DOI:** 10.3390/cancers16101851

**Published:** 2024-05-12

**Authors:** Julian Fromm, Alexander Klein, Franziska Mentrup, Lars H. Lindner, Silke Nachbichler, Boris Michael Holzapfel, Sophia Samira Goller, Thomas Knösel, Hans Roland Dürr

**Affiliations:** 1Department of Orthopaedics and Trauma Surgery, Orthopaedic Oncology, Musculoskeletal University Center Munich (MUM), LMU University Hospital, LMU Munich, D-81377 Munich, Germany; julian.fromm@med.uni-muenchen.de (J.F.); alexander.klein@med.uni-muenchen.de (A.K.); fmentrupfm@googlemail.com (F.M.); boris.holzapfel@med.uni-muenchen.de (B.M.H.); 2SarKUM, Center of Bone and Soft Tissue Tumors, LMU University Hospital, LMU Munich, D-81377 Munich, Germany; lars.lindner@med.uni-muenchen.de (L.H.L.); silke.nachbichler@med.uni-muenchen.de (S.N.); sophia.goller@med.uni-muenchen.de (S.S.G.); thomas.knoesel@med.uni-muenchen.de (T.K.); 3Department of Medicine III, LMU University Hospital, LMU Munich, D-81377 Munich, Germany; 4Department of Radiation Oncology, LMU University Hospital, LMU Munich, D-81377 Munich, Germany; 5Department of Radiology, LMU University Hospital, LMU Munich, D-81377 Munich, Germany; 6Institute of Pathology, LMU Munich, D-81377 Munich, Germany

**Keywords:** sarcoma, surgery, recurrence, margins, prognosis, local recurrence, re-excision, unplanned excision, predictive factor

## Abstract

**Simple Summary:**

In soft tissue sarcomas, unplanned resections, so-called Whoops procedures, do occur frequently. Whether re-resection reduces local recurrence or improves overall survival is unclear. 185 patients who underwent unplanned resection were included. Group A (n = 156) underwent re-excision, while Group B (n = 29) not. Residual tumor was observed in 60% of the resected tumors. In Group A, 8% of the patients developed local recurrence, in Group B 14% (n.s.). Overall survival and local recurrence-free survival were not different between the groups. However, within the subgroup of patients with residual disease in the re-resected specimen, survival was compromised, and the local recurrence rate was higher. Particularly for low-grade lesions, more patients could be treated without re-resections.

**Abstract:**

Background: In soft tissue sarcomas, unplanned resections, or so-called Whoops procedures, do occur quite frequently, thus primarily owing to the abundant presence of benign lesions. Whether re-resection reduces local recurrence or improves overall survival remains a topic of ongoing debate. The principle objective of this study was to analyze the outcomes of patients with soft tissue sarcomas of the extremities or trunk wall after an incidental marginal resection by comparing re-resections to individuals who declined the procedure. Methods: A total of 185 patients who underwent unplanned resection were included. These patients were stratified into two groups: Group A (n = 156) underwent re-excision, while Group B (n = 29) was treated conservatively. Depending on the clinical scenario, radio- or chemotherapy was either administered in a neoadjuvant or an adjuvant setting. The presence of residual tumor and metastatic disease was documented. Clinical outcomes, specifically local recurrence (LR), local recurrence-free survival (LRFS) and overall survival (OS), were utilized for evaluation. Results: Group B exhibited significantly larger tumors (*p* < 0.0001) and a higher mean age than Group A. Among the patients in Group A, 11 (5.9%) had contaminated resection margins (R1), and residual disease (RD) was observed in 93 (59.6%) of the resected specimens. In group B, 10 patients received adjuvant radiotherapy alone, 5 received chemotherapy alone, and 13 underwent a combined approach consisting of both radio- and chemotherapy. In Group A, 8% (n = 12) of the patients developed local recurrence (LR) during the observation period. Conversely, in Group B, this amount was 14% (n = 4) (n.s.). Of the 12 LR in Group A, 10 were found in the subgroup with residual disease. Overall survival and local recurrence-free survival were not significantly different between the groups. A total of 15% (n = 24) of the patients in Group A developed metastatic disease, while 10% (n = 3) in Group B developed metastatic disease (n.s.). Conclusions: Following the reresection of unplanned resected STS, there was no statistically significant difference observed in overall survival or LR compared to patients who did not undergo re-resection. However, within the subgroup of patients with residual disease in the re-resected specimen, the OS was compromised, and the LR rate was higher. Particularly for low-grade lesions, adopting a more conservative approach seems to be justified.

## 1. Introduction

Soft tissue sarcomas encompass a heterogeneous group of mesenchymal tumors with more than 150 distinct histological and molecular subtypes, with each exhibiting varying clinical behaviors. These malignancies constitute less than 1% of all adult malignancies [1]. Their occurrence is pervasive throughout the body, with a predilection for the extremities and the trunk [2]. The mortality rate associated with STS is as high as 40%, and distant metastasis is common [3]. Wide surgical resection with clear margins is the therapeutic gold standard, which is often combined with neoadjuvant chemotherapy, radiotherapy, or both, depending on the size, grading, and location of the tumor [3].

In many cases, an inadvertent resection, characterized by a hasty excision lacking clear margins from the surrounding tissues (“Whoops” procedure), results in an incidental R1 resection. As demonstrated by a nationwide survey in the Netherlands, where unplanned resections accounted for 18.2% of all initial surgical interventions for STS, this scenario is not uncommon. In another study from Austria analyzing patients treated in three major tumor centers, the rate of unplanned resections was even higher at 38.6% [4,5]. For such cases, the optimal therapy entails re-excision with or without adjuvant radiotherapy [6]. However, whether re-resection does indeed prolong overall survival remains a subject of debate, and several authors even recommend postponing re-excision until local recurrence might occur [7]. Others contend that re-resection yields even better results than a one-stage tumor resection [6]. One of the major problems in resolving this quandary is the absence of high-quality randomized studies. Given the inherent nature of this issue, the likelihood of conducting such investigations is very low. Consequently, a control group comparing the results of re-excision to a more conservative approach is missing in most of the studies.

The objective of this study was therefore to compare the outcomes of sarcoma patients after unplanned marginal resection and subsequent patients who declined this secondary surgical intervention for various reasons.

## 2. Methods

Between 2012 and 2021, a total of 185 patients with unplanned resection of a soft tissue sarcoma were treated at our institution. Of those, 156 individuals underwent re-resection (Group A). Twenty-nine patients opted not to undergo reresection as per their own decision (Group B).

The diagnosis was confirmed through histology of the resected or excised specimen. Before arriving at a final decision regarding subsequent surgery, radiotherapy or chemotherapy, magnetic resonance imaging (MRI) and, in some cases, computed tomography (CT) were employed to define the dimension and location of the tumor or the tumor bed. Additionally, a chest CT scan was obtained to ascertain the presence or absence of metastatic disease.

### 2.1. Surgery

All surgeries were performed by two experienced surgeons. The primary objective of surgery was to attain R0 resection whenever possible. In cases requiring it, local or free flaps were utilized to optimize surgical outcomes.

### 2.2. Radiotherapy

Radiotherapy was employed either in a neoadjuvant or adjuvant setting. The utilization and timing of radiotherapy were individually and interdisciplinarily discussed during routine tumor board meetings. In Group A n = 90 (58%) received radiotherapy; in Group B n = 23 (79%) received radiotherapy.

### 2.3. Chemotherapy

Chemotherapy, in the majority of cases, was scheduled as a combined neoadjuvant and adjuvant multiagent therapy. Typically, for soft tissue sarcomas, this regimen consisted mostly of AI (adriamycin and ifosfamide) or EIA (etoposide, ifosamid, and adriamycin), which was occasionally augmented by other protocols, including local hyperthermia in specific instances [8]. The utilization and timing of chemotherapy were also individually and interdisciplinary discussed during routine tumor board meetings.

### 2.4. Statistical Analysis

All patients underwent thorough monitoring for signs of local recurrence (LR) or distant metastasis primarily through regional MRI scans and chest radiographs. Clinical endpoints were local recurrence (LR), local recurrence-free survival (LRFS) and overall survival (OS). LRFS and OS were defined either as the duration from surgery to the first recurrence or to the event of death from any cause. For statistical analysis, overall and local recurrence-free survival were calculated according to the Kaplan-Meier method. Significance was determined using the log-rank test, the chi-square test, or the Cox proportional hazards regression model. A *p* value of less than 0.05 was considered statistically significant. The data analysis software used were MedCalc^®^ (MedCalc Software, Ostend, Belgium, 22.001) and SPSS (Version 28).

## 3. Results

A total of 185 patients with previously performed unplanned resection STS were divided into two groups depending on the planned treatment after primary incomplete resection. Group A consisted of 156 patients who underwent R1 resection, with 79 female and 77 male patients (50.6% vs. 49.4%). The mean age was 57 years (13–93 years). The predominant diagnosis was undifferentiated pleomorphic sarcoma (UPS), which was observed in 28% of the cases, followed by myxofibrosarcoma (17%), leiomyosarcoma (12%), and liposarcoma (10%). The most common location was the upper thigh in 30% of cases, followed by the lower leg in 15% and the forearm in 12%. In terms of tumor depth, 36% of the lesions were superficial, while 64% were subfascial. Tumor grading revealed 39% of the lesions to be classified as grade 3, 35% as grade 2, and 15% as grade 1. Notably, 11% of the cases lacked grading information in the (external) pathological report (Table 1). Among the patients in Group A, the initial tumor size could be evaluated in 114 out of 185 cases, while in Group B, this information was available for 25 out of 29 patients. Group B exhibited significantly larger tumors (*p* < 0.0001).

In all 156 cases, patients underwent incomplete primary resection (R1), thus necessitating subsequent wide reresection or initiating neoadjuvant radiotherapy less than 3 months later at our institution. Notably, for 11 patients in Group A (5.9%), the resection margin was contaminated (R1). Residual disease (RD) was observed in 93 (59.6%) of the resected specimens.

In Group B (n = 29), 14 of the patients were female, and 15 were male. The mean age was 60 years (31–85 years). The most prevalent diagnosis was UPS in 28% of the cases, followed by liposarcoma in 24% and malignant peripheral nerve sheath tumor (MPNST) in 10% of the cases. The predominant location was the upper thigh in 38% of the cases, followed by the trunk in 20% and the pelvis and lower leg in 10% of the cases. The subfascial location was observed in 97% of the cases, while the superficial location was documented in 3% of the cases. Thirty-nine percent of the patients exhibited a grade 3 lesion, 35% exhibited a grade 2 lesion, and 15% exhibited a grade 1 lesion. Notably, in 12% of cases, the pathological report (external) lacked grading (Table 1). Among the 29 patients, 9 received adjuvant radiotherapy alone, 5 received chemotherapy alone, and 14 received both radio- and chemotherapy.

### Local Recurrence and Survival

The surviving patients’ median follow-up was 40 months in Group A and 43 months in Group B. In Group A, n = 17 (10.9%) and, in Group B, n = 1 (3.8%), had a follow-up in less than 12 months.

In Group A, 8% (n = 12) of the patients experienced local recurrence (LR) during the designated observation period. In comparison, in Group B, 14% (n = 4) of patients exhibited LR (Figure 1), which was a difference deemed statistically not significant (n.s.). Within Group A, 10 out of 12 patients experiencing LR had residual disease (10.8% LR), whereas only two LR cases were observed in patients without RD (3.2%) (n.s.). The overall survival and local recurrence-free survival were not different between the groups (Figure 1 and Figure 2). But in those patients having residual disease, the overall survival was reduced (Figure 3). Radiotherapy did not significantly change the results (Figure 4a,b).

Dermatofibrosarcomas are known to have an excellent outcome in terms of LR and OS, which might introduce bias comparing between the two groups. In Group B, no DFSP had been included. In Group A, 13 patients with DFSP showed no LR, no metastatic disease, and no deaths. Excluding DFSP in analysis of the LR and OS, the results did not change.

Fifteen percent (n = 24) of the patients in Group A developed metastatic disease (MD), while in Group B, 10% (n = 3) suffered from MD (n.s.). In Group A, 16% (n = 25) of patients, and in Group B 24% (n = 7) of patients, died during the observation period (n.s.).

## 4. Illustrating Case

A 28-year-old female complained of pain and swelling in the left lateral hindfoot for 6 years, with the pain in the last year slowly growing. The lesion was resected 5 months after MRI investigation (Figure 5a,b). Histology showed a monophasic synovial sarcoma of about 2 cm. Formal grading could not be obtained, but, regarding the numbers of mitotic figures, it was judged as G1. All margins had been contaminated. A CT scan of the thorax and abdomen could not prove any further disease. A second MRI 4 weeks after resection showed a lesion for which it was not clear whether it was residual disease or scar tissue. In the interdisciplinary board meeting, we decided to go for neoadjuvant radiotherapy followed by re-resection and plastic reconstruction. Radiotherapy started 12 weeks after the first surgery. A total of 50 Gy in fractions of 2 Gy were applied. Re-resection was performed 40 days after radiotherapy. For coverage of the defect, a free muscle flap (gracilis) and a skin graft were used. Histology determined residual tumor infiltrating the nerve sheaths and the nerve along 2.4 cm with a clear margin of at least 8 mm. Flap revision with thinning was performed two years later; the patient is free of any tumor 3 years after re-resection (Figure 5c).

## 5. Discussion

Both study groups demonstrated tumors with a comparable histological grading, type of lesions, and patient age. Notably, in the patients who were not re-resected, the number of deep lesions was higher, and the tumor size was significantly larger. We attributed this observation to the imperative for more extensive re-resections, which could have influenced patients to opt for a nonsurgical approach. Consequently, both groups displayed heterogeneity, with more unfavorable lesions in Group B.

When benchmarked against existing studies, our local recurrence rate of 8% and OS in the re-excision group align closely with reported figures [9]. It is worth highlighting that, in contrast to other studies, we included a control group of non-resected lesions. Many previous studies employed primary resected STS as their control group, thus ultimately leading to the conclusion that there are no prognostic differences between re-resected and primary resected lesions [9,10].

In a multinational survey including 697 macroscopically complete yet unplanned resected lesions published in 2021, patients who underwent re-resection exhibited a significantly better overall survival ranging from 7–10% compared to those with primary resections. However, it is noteworthy that the rate of LR approximately doubled, thereby escalating from 5% to 12–15% in re-resected patients [7]. Consequently, the authors advocated for a “wait and see” approach for LR in cases of macroscopically complete R1 resections. The notion that re-resection might contribute to better overall survival than standard therapy was initially observed in the early 1980s and subsequently confirmed by other studies [6,11,12]. As such, re-resection remains the gold standard therapy whenever feasible. In a comprehensive 2008 survey from Birmingham involving the analysis of 402 unplanned resections within a cohort of 2.201 STS [13], routine re-resection was used, thus uncovering residual tumors in 47% of the cases. Among those with persistent positive margins after the re-resection of high-grade tumors, the risk of LR was 60%. It is important to note that there was no control group in this study. Nonetheless, the authors concluded that the high proportion of residual tumor justifies the practice of routine re-resection.

Residual disease has been found to be evident in approximately 23–83% of re-resections (this study 59.6%) and is notably associated with a higher rate of MD. This presence of residual disease emerges as an independent adverse prognostic factor, thus affecting both disease-specific survival and LR rates [9,10,12,14,15,16].

While re-resection may reduce the risk of LR, several studies, such as the one conducted by Rutkowski et al. focusing on liposarcoma patients, suggest that it may not influence disease-specific survival [17]. Traub et al., in their study encompassing 94 re-resected patients from the Toronto group, found no difference in the LR rates and overall survival [16]. Additionally, a study by Smolle et al. showed the outcomes of the Graz and Vienna groups [5], thus revealing that 38.6% of STS patients had undergone unplanned resection (n = 281), and in this cohort, both the OS and LR rates were not different from those of patients with primary resections. This must be seen in favor of a “wait and see” strategy. The strategy of waiting for LR and then deciding upon a secondary surgical intervention delays re-resection, thus potentially heightening the risk of MD. However, intriguingly, this delay in re-resection did not seem to have any influence on overall outcomes [18].

In 2019, a comprehensive analysis conducted by a notable group of French sarcoma experts, comprising 622 patients, was published [19]. Patients were divided into three groups based on their re-resection status: those undergoing re-resection in a specialized sarcoma center, those receiving re-resection outside such a center, and those without re-resection. Remarkably, there was no difference in the 5-year OS among these groups. Local recurrence-free survival was best in the group of patients re-resected in the sarcoma center and worst in the patients without reresection. LR was seen in 9.3%, 21.1%, and 31.9% of patients, respectively. Consequently, the authors concluded that routine re-resection offers the best local control but does not influence OS. Additionally, the authors concluded that delayed re-resection following LR might constitute a viable option.

In contrast, the same author group published another study in 2022 utilizing data from 1284 patients of the French Sarcoma Network who had undergone initial STS surgery in nonspecialized centers with surgical margins that were microscopically tumor-invested (R1 resection) [20]. Among this cohort, 1029 were further assessed during follow-up, with 698 of them undergoing re-resection and 331 opting for no additional surgical intervention. In a multivariate analysis, it was demonstrated that this significantly influenced the overall survival and LR rates. Consequently, the revised conclusion emphasized the systematic consideration of re-resection in such cases.

Investigating the timing (within or beyond 2 months after initial surgery) of re-resection in 131 patients, Sacchetti et al. found no impact on the prognosis [21]. They suggested adopting a “wait and see” strategy in patients with low-grade tumors. Moreover, Sacchetti et al. conducted a metaanalysis in 2021, encompassing 32 articles [22], to compare the outcomes of patients who underwent re-resection versus planned resection, thus demonstrating no elevated risk for a higher LR rate. Hence, they concluded that re-resection should remain the gold standard of care in these patients. However, they acknowledged that a “wait and see” approach could be justified in cases of unplanned excisions of low-grade lesions under certain circumstances.

Studies focusing on one entity in high-risk lesions are rare. In a cohort of 109 patients with synovial sarcoma and unplanned resection needing re-resection, the authors categorized the patients into three distinct groups: those with no residual disease, those with residual disease, and those experiencing LR [23]. Analysis of both groups with tumors that had been immediately re-resected revealed no difference in terms of the OS and LR rate. However, the group of patients undergoing delayed resection subsequent to LR exhibited a notably poorer prognosis with respect to both the OS and LR rates. Consequently, the authors concluded that a “wait and see” approach in synovial sarcoma with unplanned resection should be avoided.

Nonetheless, one must be aware that re-resections entail a higher risk of necessitating flap coverage or skin grafting when compared to primary resections [12,16,24]. This risk is particularly elevated in tumors of the distal extremities, including the hand and foot [25]. For instance, at the MD Anderson Cancer Center in Texas, among 67 patients undergoing re-resection, 73% of patients needed plastic surgery, and 45% developed wound complications [26].

Pertinent to any discussion is the additional financial burden of re-excision [27].

In the authors’ opinion, it is imperative to engage in a comprehensive discussion with the patient regarding these findings. A re-resection procedure, if uncomplicated, holds promise to improve the OS and lower the risk of LR. However, in cases where the anatomic location (i.e., pelvis, spine) poses difficulties, this potentially compromises limb function due to mutilating surgical procedures. A routine re-resection should be avoided if no gross residual disease is observed. Instead, radiotherapy should at least be part of the therapeutic regimen in non-re-resected tumors given its significant impact on both overall survival and progression-free survival [28,29]. Furthermore, for patients who have undergone re-resection, the LR was significantly influenced by the administration of additional radiation therapy, but notably, overall survival remained unaffected [30].

Interestingly, our group of patients with no re-resection radiotherapy showed a trend towards a higher rate of LR. We think that is a bias. Based on the clinical decision, those six patients were considered to have a low risk of LR and did therefore not receive RTX.

Outlined in a contemporary current concept review, some general considerations were stressed [31]. Educational surgical programs aimed at averting unplanned resections should be established. Re-resection is recommended in the majority of cases but entails a higher probability of necessitating reconstructive procedures. Unplanned resections are associated with a higher rate of LR. Adjuvant radiotherapy does not mitigate the risk of LR but is used in a manner akin to planned resections.

To propose re-resection as the primary course of action in a majority of cases is the safest recommendation from an oncological perspective. However, considering the often-necessary extensive surgical procedures, the heightened rate of complications observed in many cases of multimodal treatment, and the body of literature discussed earlier, advocating for a more conservative approach seems to be justified, especially in cases of low-grade lesions.

There is no doubt that the diagnosis and therapy of STS are best performed within a specialized sarcoma center. However, a notable challenge arises from the fact that STS constitutes a mere fraction of tumors, with approximately 1 observed in every 300 soft tissue tumors. Consequently, these centers do not have the capacity and are not designed to handle the plethora of classic benign soft tissue lesions. Therefore, prioritizing the education of surgeons outside these specialized sarcoma centers and implementing guidelines to avoid marginal resections without prior biopsy in cases of deep or sizable soft tissue tumors is of utmost importance [32,33]. Appropriate imaging prior to surgery is the first important step. This alone improved the outcome of our patients [34].

## 6. Limitations of the Study

This study is retrospective in nature. The patients had been registered prospectively, as were the pathological assessments. The inclusion criteria encompassed patients not only diagnosed with sarcoma but those with all subcategories. Our control group of non-resected patients is small, thus presenting an inherent bias particularly toward larger tumors in older patients and consequently predisposing them to a worse prognosis. However, it is noteworthy that ultimately, there was no significant prognostic disadvantage in the cohort of patients who underwent re-resections.

Accounting for the high numbers of patients after “Whoops” procedures even seen in our own institution every year, we thought of a randomized study between re-excision and not undergoing re-excision. But, accounting for the high number of residual disease in Group A, we think it is difficult to establish a randomization without being responsible for the absence of re-resection in more than 50% patients with residual disease.

## 7. Conclusions

Following the re-resection of unplanned resected STS, the overall survival was not significantly different when compared to a group of non-re-resected tumors. Likewise, there was no significant difference in terms of LR. However, within the subgroup of patients with residual disease in the re-resected specimen, both the OS and LR demonstrated a less favorable outcome. Consequently, engaging in a thorough discussion with the patient regarding the option of re-resection, including its cons and pros, is imperative. Especially in low-grade lesions, a more conservative approach might be justified.

## Figures and Tables

**Figure 1 cancers-16-01851-f001:**
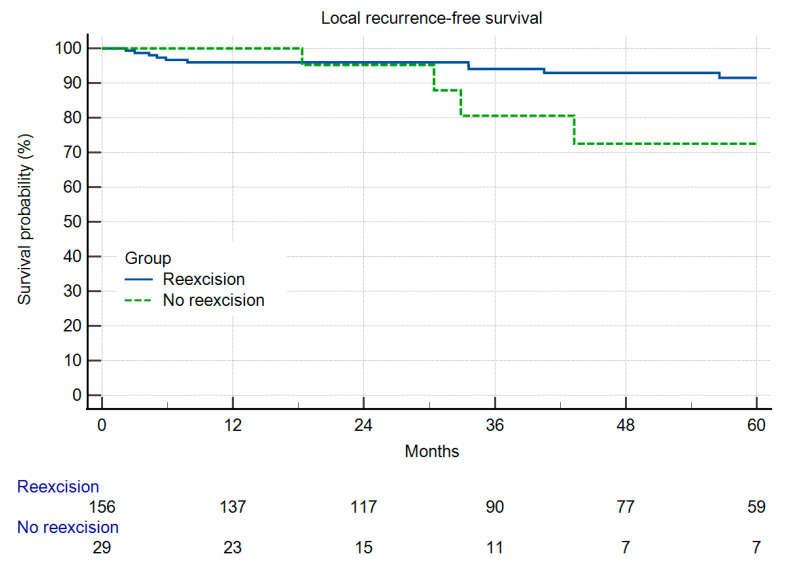
Local recurrence-free survival in 156 patients with and 29 patients without re-resection (*p* = 0.2696).

**Figure 2 cancers-16-01851-f002:**
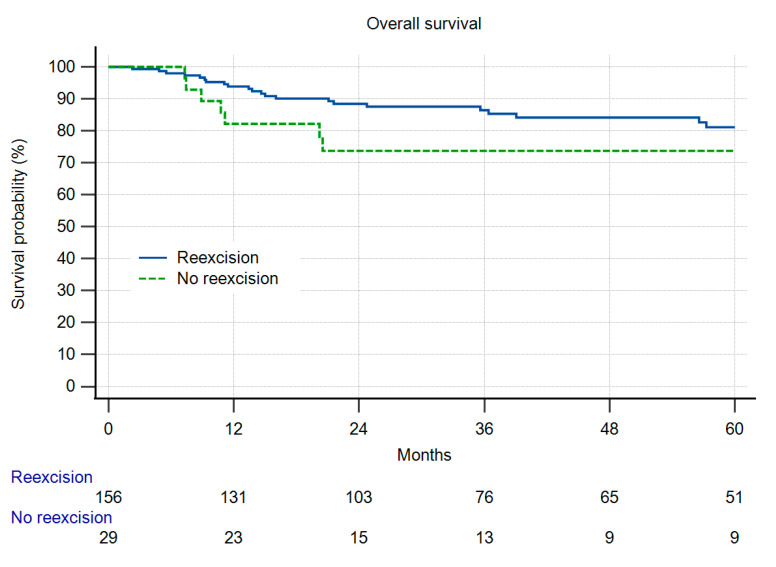
Overall survival in 156 patients with and 29 patients without re-resection (*p* = 0.2937).

**Figure 3 cancers-16-01851-f003:**
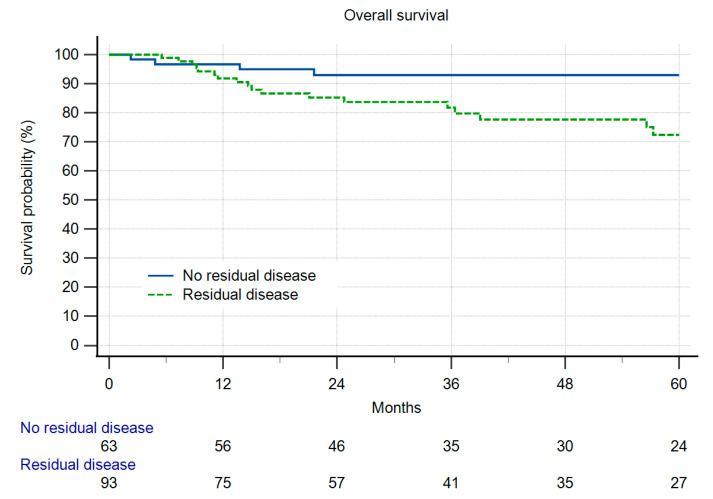
Overall survival in 156 patients after re-resection with or without residual disease in the resected specimen. (*p* = 0.0466).

**Figure 4 cancers-16-01851-f004:**
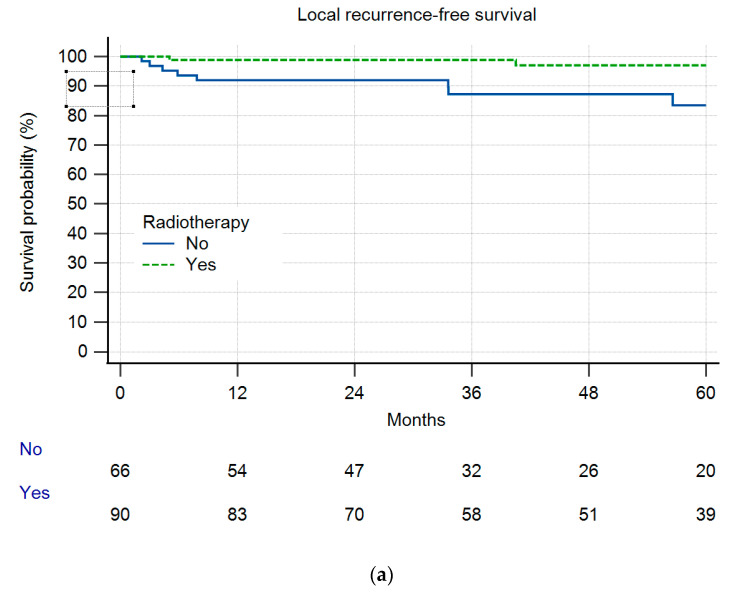
(**a**): Local recurrence-free survival in 156 patients after re-resection with or without radiotherapy (*p* = 0.1131). (**b**): Local recurrence-free survival in 29 patients after re-resection with or without radiotherapy (*p* = 0.5320).

**Figure 5 cancers-16-01851-f005:**
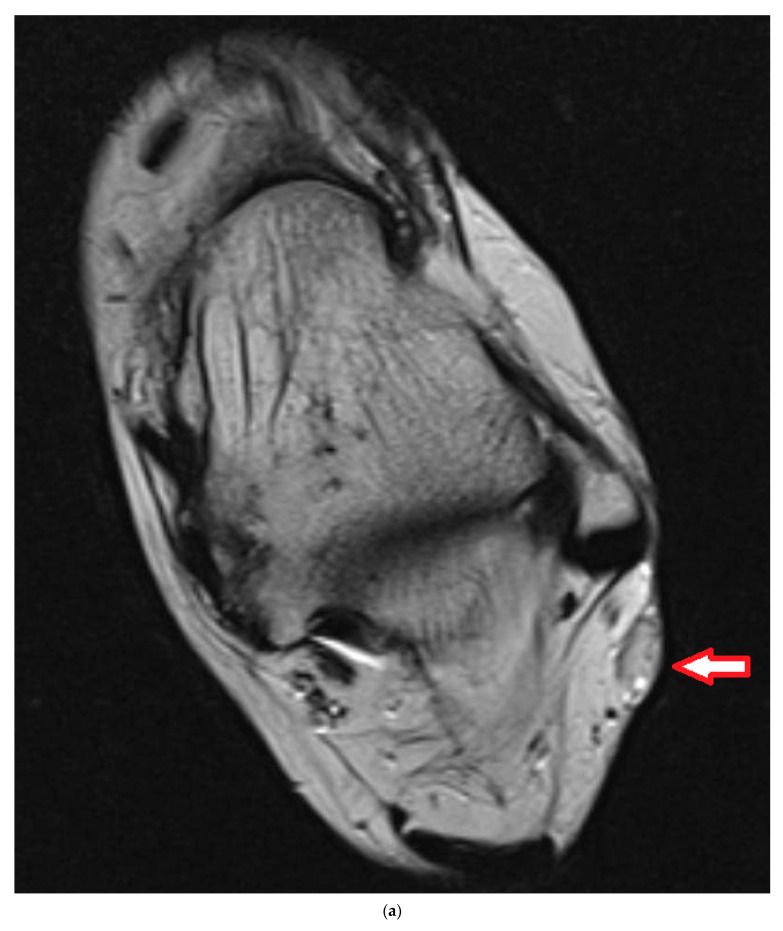
(**a**): Axial MRI (T2tseR) showing a subcutaneous tumor at left hindfoot of about 2 cm size. The lesion was judged as a benign soft tissue tumor. (**b**): Coronal MRI (T1) showing the subcutaneous extension. (**c**): Clinical situation before thinning of the flap.

**Table 1 cancers-16-01851-t001:** Details of tumor characteristics of both groups.

	Reresection	No Reresection
**Grading**		
G1	23 (14.8%)	6 (24%)
G2	54 (34.8%)	8 (32%)
G3	60 (38.7%)	11 (44%)
X	19 (12.2%)	4 (13.8%)
**Mean age**	57 ys	60 ys
**Location**		
Superficial	56 (35.9%)	1 (3.4%)
Deep	100 (64.1%)	28 (96.6%)
**Size of the lesion**		
Mean size (cm)	4.5	10.1
**Entity**		
UPS	43 (27.6%)	8 (27.6%)
Myxofibrosarcoma	27 (17.3%)	2 (6.9%)
Leiomyosarcoma	19 (12.2%)	2 (6.9%)
Liposarcoma	16 (10.3%)	7 (24.1%)
Dermatofibrosarcoma	13 (8.3%)	
Synovialsarcoma	11 (7.1%)	2 (6.9%)
MPNST		3 (10.3%)
Others	27 (17.3%)	5 (17.2%)
**Radiotherapy**	90 (57.7%)	23 (79.3%)

## Data Availability

The datasets used and analyzed during the current study are available from the corresponding author upon reasonable request.

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
