# Peer review of "Unplanned Resections of Soft Tissue Sarcomas—Necessity of Re-Resection?"

_cancers, 2024, doi:10.3390/cancers16101851_

Round 1
Reviewer 1 Report
Comments and Suggestions for Authors
This retrospective analysis of 185 patients presented in a tertiary referral center for sarcoma adds some insight to the significant question whether re-excision in sarcoma is important for the outcome in terms of the risk of local recurrence and overall survival.
In my opinion, it is essential to add information on adjuvant radiotherapy before drawing any conclusions about local control in sarcoma treatment. Data on adjuvant radiotherapy are only presented for group B. I think this information should be added to table 1 and could potentially be a big confounder in the present analysis.
I am also missing details on the duration of follow-up in both groups, it should be presented and shown to be similar in both groups.
It was intriguing to read that in 59,6% of the patients in group A residual disease was found and I wondered how that correlated to the MRI scan that was performed (Pag 2, 35). Was it already clear that a residual tumor was present, or was it a true incidental finding?
It might be better to withdraw the group of dermatofibrosarcoma from the analysis since they are known to have an excellent outcome in terms of overall survival, which might introduce bias comparing between the two groups.
Considering Figure 1. Please add p value since there is a clear difference between the two groups. This is actually also the case for figure 2. These analysis might not give a significant p-value, which is probably due to the small number of patients in group B, however might still be a relevant finding. I would suggest to ask for a statistical review or a collaboration with a statistician to make the right interpretation of the analysis. Right interpretation of these data might also slightly change your conclusions on the safety for omitting re-excision in these patients.
It would be interesting to know what were the reasons for the patients to choose to omit re-excision. Could this be retrieved from the patient files?
The authors state that there is a larger need for flap coverage or skin grafting when performing re-excision (pag7, 17). How does this hold for their population? This information could also be added to table 1.
I don’t see the need of summarizing the six crucial considerations of paper [31] in the discussion.
I think it should be noted in the limitation section, that true prospective data are needed to draw a sound conclusion on the safety of omitting re-excisions in soft tissue sarcoma patients.
Comments on the Quality of English Languagepag 7, 40 resections
Author Response
Reviewer #1:
- In my opinion, it is essential to add information on adjuvant radiotherapy before drawing any conclusions about local control in sarcoma treatment. Data on adjuvant radiotherapy are only presented for group B. I think this information should be added to table 1 and could potentially be a big confounder in the present analysis.
This is a very important point we missed in the manuscript. We made the changes as follows. In the results section, paragraph radiotherapy, the sentence:
“In group A n=90 (58%), in group B n=23 (79%) received radiotherapy.”
was added. In addition we added that information in table 1.
We added two more figures (4a and b) showing that radiotherapy had only a trend to reduce LR in group A (4a) and was a (n.s.) negative factor in group B (4b). We think this is a bias. For that we added in the discussion:
“Interestingly in our group of patients with no reresection radiotherapy showed a trend towards a higher rate of LR. We think that is a bias. Based on the clinical decision those 6 patients were considered to have a low risk of LR and did therefore not receive RTX.”
In the results section
“Radiotherapy did not significantly change the results (Fig. 4A,B).”
was added.
- I am also missing details on the duration of follow-up in both groups, it should be presented and shown to be similar in both groups.
This is good clinical practice and should have been done already in the first version of this manuscript. We added in the results section paragraph “Local recurrence and survival” the following:
“In surviving patients` median follow-up was 40 months in group A and 43 months in group B. In group A, n=17 (10.9%) and in group B, n=1 (3.8%) had a follow-up less than 12 months.”
- It was intriguing to read that in 59,6% of the patients in group A residual disease was found and I wondered how that correlated to the MRI scan that was performed (Pag 2, 35). Was it already clear that a residual tumor was present, or was it a true incidental finding?
The reviewer is right. As stated in the discussion this finding is confirmed also by other studies. There had been some cases there MRI did indicate residual disease but in most of the patients the discrimination between scare tissue and residual disease in MRI was not possible. Mainly because of the short period of time after first surgery. We did some investigations on that (Goller SS, Reidler P, Rudolph J, Rückel J, Hesse N, Schmidt VF, Dürr HR, Klein A, Lindner LH, Di Gioia D, Kuhn I, Ricke J, Erber B. Impact of postoperative baseline MRI on diagnostic confidence and performance in detecting local recurrence of soft-tissue sarcoma of the limb. Skeletal Radiol. 2023 Oct;52(10):1987-1995.) showing that even with a baseline MRI this is difficult to asses.
- It might be better to withdraw the group of dermatofibrosarcoma from the analysis since they are known to have an excellent outcome in terms of overall survival, which might introduce bias comparing between the two groups.
In group B no DFSP had been included. In group A 13 patients with DFSP showed no LR, no metastatic disease and no deaths. This underlines the comment of the reviewer. We did a recalculation for LR and OS. This did neither change the graphs nor the significance level. But this comment of the reviewer should be included in manuscript. We added:
“Dermatofibrosarcomas are known to have an excellent outcome in terms of LR and OS overall survival, which might introduce bias comparing between the two groups. In group B no DFSP had been included. In group A 13 patients with DFSP showed no LR, no metastatic disease and no deaths. Excluding DFSP in analysis of LR and OS, the results did not change.”
- Considering Figure 1. Please add p value since there is a clear difference between the two groups. This is actually also the case for figure 2. These analysis might not give a significant p-value, which is probably due to the small number of patients in group B, however might still be a relevant finding.
Done as proposed.
- I would suggest to ask for a statistical review or a collaboration with a statistician to make the right interpretation of the analysis. Right interpretation of these data might also slightly change your conclusions on the safety for omitting re-excision in these patients.
We did that. In short she said: “Give me more data especially a larger number of patients in group 2 then we might come up with a more meaningful analysis”. And “The biggest issue in criticism of this study is the bias of the group building itself”. We mentioned that in the “Limitations” section at the end. That leads to another point of the reviewer.
- I think it should be noted in the limitation section, that true prospective data are needed to draw a sound conclusion on the safety of omitting re-excisions in soft tissue sarcoma patients.
Yes, the reviewer is right. Accounting for the high numbers of patients after “Whoops” procedures even seen in our own institution every year, we thought of an easy to establish randomized monocentric study between re-excision and not re-excision. But then we noticed high number of residual disease in group A. We think it is difficult to establish a randomization being responsible for not reresection of more than 50% residual disease. But this needs to be mentioned as commented by the reviewer.
We added to the limitations section:
“Accounting for the high numbers of patients after “Whoops” procedures even seen in our own institution every year, we thought of a randomized study between re-excision and not re-excision. But accounting for the high number of residual dissease in group A, we think it is difficult to establish a randomization being responsible for not re-resection of more than 50% patients with residual disease.”
- It would be interesting to know what were the reasons for the patients to choose to omit re-excision. Could this be retrieved from the patient files?
It was very difficult for us to establish this group. In our own files we do have only a very small group of patients who choose not to undergo a second surgery. The main reason for that was the necessity for a “larger” intervention including plastic surgery or the age of the patients letting them opt against any larger surgery. But most of the patients in this group are collected from the files of the departments of radiotherapy and oncology. They saw those patients send to them by external institutions or physicians where this decision has already been made. For that the final reasons for deciding against a second surgery are not very well documented.
- The authors state that there is a larger need for flap coverage or skin grafting when performing re-excision (pag7, 17). How does this hold for their population? This information could also be added to table 1.
In the data analysis up to now flap coverage or skin grafting was not included in the factors evaluated. We could of course gain that information by going through the files again. By writing this lines I asked our first author who did the data acquisition to do just that. But this will need some time but the files are digitalized that makes it easier. Reviewer 2 asked for a representative case study. So I will do that now choosing a case with the necessity of flap coverage. If the revision(s) will take more time, we would then add the numbers of flap coverage and/or skin grafting as proposed to table 1. We hope that this is sufficient for you.
- I don’t see the need of summarizing the six crucial considerations of paper [31] in the discussion.
We shortened that paragraph as follows:
“Outlined in a contemporary current concept review some general considerations were stressed [31]. Educational surgical programs aimed at averting unplanned resections should be established. Reresection are recommended in the majority of cases but entail a higher probability of necessitating reconstructive procedures. Unplanned resections are associated with a higher rate of LR. Adjuvant radiotherapy does not mitigate the risk of LR but is used in a manner akin to planned resections.”

Reviewer 2 Report
Comments and Suggestions for Authors
The authors study the Unplanned Resections of Soft Tissue Sarcomas, and ask Necessity of Reresection. The study is interesting, however I have concerns to discuss.
It is natural to look at it according to the degree of malignancy.
We want novelty.
Please find the additional comments below:
-What exactly is the conservative treatment of low-grade tumors?
-If re-excision is done, how long is the period of time allowed after the first time?
-What about functional reconstruction?
-Please summarize case details in a table.
Also, please provide a representative case study.
Author Response
Reviewer #2:
- What exactly is the conservative treatment of low-grade tumors?
In group B (conservative) all 6 patients with a G1 tumor received radiotherapy, none chemotherapy.
- If re-excision is done, how long is the period of time allowed after the first time?
This is an important question. There is no literature about that. We considered a time frame of less than 3 months for further local therapy (either surgery or radiotherapy) as adequate. In the manuscript we changed:
“In all 156 cases, patients underwent incomplete primary resection (R1), necessitating subsequent wide reresection at our institution .”
to
“In all 156 cases, patients underwent incomplete primary resection (R1), necessitating subsequent wide reresection .”
- What about functional reconstruction?
This was done as necessary. We did not evaluate the functional reconstructions and the functional results in this study. They might be very different in both groups, but this was not included in our investigation and is very much dependent to the location of the tumor. Reviewer #1 asked also:
The authors state that there is a larger need for flap coverage or skin grafting when performing re-excision (pag7, 17). How does this hold for their population? This information could also be added to table 1.
In the data analysis up to now flap coverage or skin grafting was not included in the factors evaluated. We could of course gain that information by going through the files again. By writing this lines I asked our first author who did the data acquisition to do just that. But this will need some time but the files are digitalized that makes it easier. Reviewer #2 asked for a representative case study. So I will do that now choosing a case with the necessity of flap coverage. If the revision(s) will take more time, we would then add the numbers of flap coverage and/or skin grafting as proposed to table 1.
We hope that this is also sufficient for you.
- Please summarize case details in a table.
We enlarged table 1 with details to radiotherapy. We may also provide further details if necessary but we wanted to focus on the typical and prognostic relevant aspects.
- Also, please provide a representative case study.
This is a very good idea which shows the problematic issue of reresection. We did it as proposed.

Reviewer 3 Report
Comments and Suggestions for Authors
There is only one observation I want to do concerning the article structure.
I think that only out of haste you forgot to individualize, to separate the Chapter Results from the previous Chapter - Methods.
Author Response
- I think that only out of haste you forgot to individualize, to separate the Chapter Results from the previous Chapter - Methods.
Important point. This should not have been forgotten. We introduced a chapter heading “Results”.
